# Affective Saturation Index: A Lexical Measure of Affect

**DOI:** 10.3390/e23111421

**Published:** 2021-10-28

**Authors:** Alessandro Gennaro, Valeria Carola, Cristina Ottaviani, Chiara Pesca, Arianna Palmieri, Sergio Salvatore

**Affiliations:** 1Department of Dynamic and Clinical Psychology and Health Studies, Sapienza University of Rome, 00185 Rome, Italy; a.gennaro@uniroma1.it (A.G.); valeria.carola@uniroma1.it (V.C.); chiara.pesca@uniroma1.it (C.P.); 2IRCCS Santa Lucia Foundation, 00179 Rome, Italy; cristina.ottaviani@uniroma1.it; 3Department of Psychology, Sapienza University of Rome, 00185 Rome, Italy; 4Padua Neuroscience Center, Department of Philosophy, Sociology, Education and Applied Psychology, University of Padua, 35122 Padua, Italy; arianna.palmieri@unipd.it

**Keywords:** affect, affective saturation index, meaning, text analysis, physiology, heart rate variability

## Abstract

Affect plays a major role in the individual’s daily life, driving the sensemaking of experience, psychopathological conditions, social representations of phenomena, and ways of coping with others. The characteristics of affect have been traditionally investigated through physiological, self-report, and behavioral measures. The present article proposes a text-based measure to detect affect intensity: the Affective Saturation Index (ASI). The ASI rationale and the conceptualization of affect are overviewed, and an initial validation study on the ASI’s convergent and concurrent validity is presented. Forty individuals completed a non-clinical semi-structured interview. For each interview transcript, the ASI was esteemed and compared to the individual’s physiological index of propensity to affective arousal (measured by heart rate variability (HRV)); transcript semantic complexity (measured through the Semantic Entropy Index (SEI)); and lexical syntactic complexity (measured through the Flesch–Vacca Index (FVI)). ANOVAs and bi-variate correlations estimated the size of the relationships between indexes and sample characteristics (age, gender), then a set of multiple linear regressions tested the ASI’s association with HRV, the SEI, and the FVI. Results support the ASI construct and criteria validity. The ASI proved able to detect affective saturation in interview transcripts (SEI and FVI, adjusted R^2^ = 0.428 and adjusted R^2^ = 0.241, respectively) and the way the text’s affective saturation reflected the intensity of the individual’s affective state (HRV, adjusted R^2^ = 0.428). In conclusion, although the specificity of the sample (psychology students) limits the findings’ generalizability, the ASI provides the chance to use written texts to measure affect in accordance with a dynamic approach, independent of the spatio-temporal setting in which they were produced. In doing so, the ASI provides a way to empower the empirical analysis of fields such as psychotherapy and social group dynamics.

## 1. Introduction

The measurement of the level of affective intensity is a relevant issue for both theoretical and practical reasons. Affect plays a central role in how individuals make sense of experience [1,2,3] as well as in how executive [4] and higher-order cognitive processes [5] operate. A high level of affect intensity is associated with psychopathological conditions [6,7,8] as well as maladjusted forms of behaviors, such as gambling [9,10,11,12] or academic drop-out [13]; again, the elaboration of affect and the mode of variation of its intensity within and throughout sessions is a core focus of the analysis of the psychotherapeutic process and clinical change [14,15,16,17,18]. Affect is also involved in psychosocial and socio-political phenomena. Social representations of forms of alterity have proven to frame their objects in terms of affect-laden meanings—e.g., the securitization frame, i.e., the view of alterity as an incumbent radical threat, is the most common way migration is conveyed by Western media [19]. In/out-group polarization, populism, xenophobia, hate speech, and conspiracy theories are phenomena that, though different in content and determinants, are all characterized by the role played by affect, and, more specifically, by the affect-laden friend–foe schema [20,21,22]. More generally, affective activation has been considered a major response by which individuals and social groups cope with uncertainty [23,24,25,26]; accordingly, the analysis of the impact and design of social communication (in contexts such as politics, media, but also health, education, marketing, security, urban planning, and civic engagement) benefit from understanding the capacity of the message to pander to and/or oppose the target’s affective response [27]. 

Due to the relevance of the topic to so many areas of psychological investigation, it is not surprising that numerous attempts have been made to measure the main characteristics of affect. Measures can be collected in three broad clusters: physiological, self-report, and, behavioral measures.

Physiological measures focus on both central (e.g., electroencephalography) and peripheral (e.g., electrodermal conductance, heart rate variability (HRV)) signals to estimate bodily affective activation (for a review, see [28]). The validity of these indexes is generally robust, given that they can be considered direct measures of the intensity of the body’s physiological activation. However, physiological measures are generally not easy to use, given that, in most cases, their application and computation require technical devices and skilled researchers. Moreover, they often require implementation in controlled conditions, and this limits their ecological validity.

Self-report measures skip some of these limitations. Yet, although these kinds of measures are widely used in many domains of investigation, their validity is considerably jeopardized by the subjects’ inherently low capability to reliably detect their own inner state [29,30]. This limitation is reflected in the weak association with both psychophysiological [31] and behavioral measures [32].

Behavioral measures offer an alternative to self-report. Many aspects of overt behavior—such as vocal fundamental frequency [33], speech rate [34], facial expressions [35], and whole-body posture [36]—have been proposed, conceived as markers of one or more features of the intensity of the affective state. However, these measures have been criticized because the association between behavior and affective states is not invariant but depends on contextual conditions [37,38,39,40,41,42,43]. Above all, it must be considered that it is not always possible to involve participants in individual tasks, or even that certain analyses are not based on participants as direct sources of data. For instance, in many areas of investigation—e.g., psychotherapy process, media representations of social objects (migration, Islam, COVID-19, etc.), organizational dynamics, and marketing communication—many studies cannot but be based on textual data such as verbatim transcripts of sessions, interviews, focus groups, newspaper articles, etc. Thus, the estimation of affect in research contexts of this kind requires a measure based on textual data, rather than on behavioral responses.

The present paper intends to address this need. It presents the Affective Saturation Index (ASI), a textual-based measure of the intensity of affect. To this end, we first present the ASI, the definition of affect it is grounded on, and its rationale; then, an initial validation study is reported. Three aspects of the ASI make it relevant from a dynamic systems theory standpoint. First, the ASI is based on a semiotic interpretation of affect, which is in turn grounded on a field-dynamic conception of meaning, which models it as an emergent property of sensemaking [44]. Second, dynamic systems theory informs the methodological framework on which the ASI’s computation is based (in particular, the dimensional model of meaning). Third, the ASI is designed to detect the ongoing flow of meaning-making, enabling the application of strategies for data analysis (e.g., Time Series Analysis) aimed at modeling the dynamic evolution of communication and cognitive processes.

## 2. The Affective Saturation Index

### 2.1. The Semiotic Definition of Affect at the Basis of the ASI: Affect as Meanings

The ASI is based on the view of affect as global embodied meanings [1,2,3,44,45,46]—namely, patterns of activation of the whole body that provide the individual with a global experience of the world as a totality. For instance, when a person is happy—though they may be so because of something—their sense of pleasantness radiates over and fills their whole current experience of the world. As proposed by Feldman Barrett [29] (p. 30), affect is the “neurophysiological barometer” thanks to which the body maps the ongoing, immediate coupling with the world and, in so doing, prepares itself to address its variations.

It is worth underscoring that it is this function of dynamic mapping that makes affect a *meaning*—albeit of an embodied kind. Indeed, insofar as one assumes the pragmatist view of meaning—i.e., a response triggered in the interpreter by something in order to interpret that something [47] (p. 228)—then affect is meaning because it consists precisely of a response activated in the (body of the) interpreter by something, in order to interpret that something. Thus, as Peirce claims explicitly [48], affect can be conceived as a specific type of sign—namely, signs that make sense of the world in terms of basic patterns of bodily activation [49].

This semiotic interpretation of affect is framed in, and fully consistent with, the bodily nature of the cognitive process recognized by embodied cognition theories over the last quarter-century [50,51,52,53]. Moreover, it finds further support in psychoanalytic theory [54,55] and in semiotic-oriented cultural psychology [3,55,56]. Incidentally, this view enables affect to be distinguished from emotions, where the latter are discrete inner states (e.g., anger, joy) combining the state of body activation (i.e., the affect) with its categorization, occurring in accordance with contextual cues [29].

For the current discussion, it is worth highlighting the specificity of affective meaning. Given its embodiment and globality, affect is a (a) hypergeneralized, (b) homogenizing, (c) bipolar, (d) asemantic, and (e) basic class of meaning. It refers to the relationship with the whole environment, rather than with specific objects (*hypergeneralization* [29]; from a psychoanalytic perspective, [57]). In so doing, the discrete aspects of the context are assimilated to the whole affective meaning, as in all cows being black at night (*homogenization* [58]). Moreover, the fact that affect is a hypergeneralized global pattern implies that it works in dichotomous mode, namely as the juxtaposition of opposite states—e.g., pleasantness/unpleasantness (*bipolarism* [21,59,60,61]). Again, it has to be underlined that these characteristics make affective meaning-making work differently from the rule-based rational mode of thought. Indeed, the fact that affect works as a single, global, hypergeneralized, homogenizing class of meaning entails it establishing relationships between elements in spite of semantic, logical, and functional differences—e.g., from the standpoint of affective meaning-making, that which is beautiful is good and trustworthy (*asemanticity* [62]). Finally, affect is an embodied form of the organization of experience that emerges from early ontogenetic stages, and therefore comes before, paves the way for, frames, and channels the subsequent rule-based processing of the semantic content (*basicity* [29,45,63,64]).

Several streams of thought converge in providing support to the characterization of affective meaning proposed above. First, one can refer to classical studies [65,66] that have shown that the prime effect works also when the association between the prime and the stimulus consists of the sharing of an affective value (e.g., pleasantness) in the absence of any semantic linkage. Similarly, the Emotional Categorization Theory (ECT [67]) states that the meaning-maker tends to assimilate objects that have the same emotional valence for him/her into the same class, regardless of their semantic relationships. The Homogenization of Classification Function Measure (HOCFUN [63]) is a method of measuring affective intensity, based on a generalization of the ECT; it assumes that affective meaning is not limited to assimilating objects, but also properties and qualities—i.e., propositional functions (e.g., an object which is seen as pleasant tends to also be seen as important). Accordingly, HOCFUN estimates the affective intensity in terms of the degree to which the individual in an evaluation task uses two semantically independent evaluation criteria (pleasantness and relevance) in a homogeneous way. Second, the psychoanalytic theory of the unconscious, more specifically the tradition triggered by Freud’s *Interpretation of Dreams* [68], has conceptualized the unconscious as a mode of thinking (rather than a place within the mind) characterized by a specific logic, that of the primary process ([45,59]; for a review of this tradition, see [55]). This tradition emphasizes that affective meaning-making is a mode of interpreting reality (*affective semiosis,* see [44]) that has its own specific logic (the primary process), which is different from rational, rule-based logic, but still systematic and endowed with inner consistency. Third, the copious literature on the semantic differential [69,70] has shown that semantic representation is grounded in three basic dimensions of meaning—evaluation, power, and activity. With few exceptions, these dimensions emerge systematically from hundreds of studies focused on a great many objects, adopting many different semantic scales, carried out in many cultural contexts, over more than half a century. Due to their generality and transversality, the three dimensions have been interpreted as basic affective meanings—for instance, the evaluation dimension channels the meaning of semantic scales as pleasant/unpleasant, beautiful/ugly, good/bad, making people use these scales in quite a similar way, despite their semantic differences. Fourth, the transversality and basicity of affect are further supported by the fact that affective meaning appears to be shared among cultures [71] and is active within all human languages [72]. Finally, framed by the Semiotic Cultural Psychology Theory, Salvatore and colleagues [5,21] have recently identified five global beliefs about life (symbolic universes, as defined by the authors) that are active in the cultural milieu of various European societies. As underscored by the authors, each symbolic universe is an implicit generalized meaning providing a global vision of what the world is/should be, which channels the way of feeling, thinking, and acting of those who identify with it. Like the semantic differential framework, symbolic universes have been considered forms of affective meaning, because each of them comprises a set of beliefs lacking semantic linkage, while associated with each other by reason of the same basic affective values (e.g., the connotation of the world as an enemy one must protect oneself from) [21].

### 2.2. ASI Rationale

#### 2.2.1. Saturation and Intensity

The ASI is designed to detect the contribution of affective meanings—as defined above: hypergeneralized, homogenizing, bipolar, asemantic, and basic classes of meaning—to the whole semantic content of the text under investigation (e.g., individual interviews, focus groups, newspaper articles, or verbatim transcripts of psychotherapy sessions). The ASI calls the extent of this contribution *affective saturation*—the more the affective meaning contributes to the whole text’s meaning, the greater the affective saturation of the text. Thus, affective saturation is like the chromatic saturation of an image—the more a given color contributes to the image, the more that image is saturated by that color.

The ASI assumes that the saturation reflects, at the level of textual output, the intensity of the meaning-maker’s affective state, associated with and influencing the cognitive process underlying the production of the text. The higher the intensity of the affective state, the more power it has to influence the meaning-making, and therefore the greater the affective saturation of the meaning-making’s textual output (see [44,64,73] for a discussion of this tenet in the context of the analysis of the psychotherapy process). In terms of the previous analogy, if an image is saturated by a given color, this is expected to be because that color plays a major role in the painter’s aesthetic desire and taste.

In summary, the ASI assumes that the affective state promotes affective saturation as a function of its intensity. On this basis, the ASI considers affective saturation the textual marker of the level of affective intensity that characterizes the meaning-maker during the production of the text.

#### 2.2.2. A Geometric Model of Affective Saturation

The ASI’s estimation of affective saturation is based on a geometrical model of meaning [45]. According to this view, meaning (e.g., an attribute, a belief, or a representation) can be modeled as a point in a semiotic space—the *Phase Space of Meaning [PSM]*, as modeled by Venuleo and colleagues [74]. The PSM is made up of dimensions, each of which maps a relevant component of environmental variability. Accordingly, the meaning is represented by the coordinates of the corresponding point in the space. For instance, the meaning of “orange” could be represented as the point in the semiotic space composed of the dimensions: form, color, and function, and with the coordinates: almost spherical, orange, food. To give a topical example, Salvatore and colleagues [20] found that the five symbolic universes they identified are framed within a semiotic space composed of three dimensions: (1) connotation of the world (polarities: friend vs. foe); (2) direction of the desire (polarities: passivity vs. engagement); and (3) form of the demand (demand for systemic resources vs. demand for community identity). Incidentally, one can see that the first two of these dimensions overlap, similar to many dimensions emerging from the semantic differential [63], providing further evidence as to the basicity of affective meaning.

Two core points of the PSM are relevant here (for a theoretical discussion, cf. [45]; for a computational model, see [6]). First, PSM dimensionality requires modulation as a function of the contextual conditions. Indeed, in any circumstance, most of the environmental sources of variation—therefore of the PSM dimensions—are not relevant, and therefore must be backgrounded to enable the cognitive system to make the relevant information pertinent and thus process it. For instance, when driving, one should focus on the aspects of the car and road that are relevant to the regulation of the vehicle, while other non-pertinent aspects should be backgrounded as dangerous sources of control loss.

Second, the PSM dimensions can be divided into two broad classes—primary and secondary dimensions. The former consists of affective meanings, each of which corresponds to a specific semantic component that maps a given quality/property of the environment, to achieve the fine attunement with the environment needed for action to be regulated. As discussed above, the affective dimensions are basic; that is, they tend to be stable, both within cultures and between individuals—namely, they work as the grounds of any meaning-making process, regardless of contextual conditions. This means that the modulation of PSM dimensionality is due to secondary dimensions, that is, to the semantic components that are activated in order to elaborate the pockets of information required to regulate action in contingent circumstances.

The above points give rise to the view of affective saturation as *the relative weight of the primary over the secondary dimensions within the PSM.* A PSM that is highly saturated by primary dimensions models a text whose global meaning is largely defined by basic affective meanings, with little room left for further semantic components. For instance, various media and political discourses provide a representation of the concept of migration almost completely saturated by the securitization frame, an expression of the friend–foe affective meaning [21], with very little room for any other meaning. Thus, framed in the geometric model of meaning, the ASI computes affective saturation in terms of the contribution of the affective meanings to the text compared to the other semantic components.

The geometric view of meaning and the related definition of affective saturation are supported by growing evidence, even if this is mostly indirect. At the level of the individual, low dimensional semiotic spaces have proven to be associated with proxies of affective activation—e.g., higher response speed to evaluation tasks and a tendency to characterize semantically distinct social objects in homogeneous ways [63]; a high need for closure and a highly negative attitude towards foreigners [21]; and a lower tendency to explore the marginal area of the attentional perception field [63]. Finally, the geometrical model of meaning outlined above was recently used by some of the present authors to frame a computational model of the form of meaning-making underpinning psychopathology—the Harmonium Model [6]. In a subsequent work, the Harmonium Model was tested by means of a simulation based on a neural network deep learning procedure. Findings showed that the neural networks simulating psychopathological meaning-making were characterized by a lower dimensional micro-dynamic compared to the neural networks simulating non-psychopathological meaning-making. At a psycho-social level, symbolic universes whose content was characterized by connotations of the social world that were positively (very high trust in people and the future, very high satisfaction, or idealization of interpersonal bonds) or negatively (very high distrust, fatalism, or rejection of rules) polarized, reactive, and generalized—in that sense affect-laden—showed a lower dimensionality than symbolic universes whose content was characterized by moderate, differentiated beliefs [20].

### 2.3. Relation to Other Text-Based Measures of Affect

The ASI shares the interest of other measures in the measurement of affect in text. The Therapeutic Cycle Model [16] focuses on the interplay of two lexical measures, based on vocabularies estimating the Emotional Tone (ET)—i.e., the affective level of the text, as detected by the use of emotion-laden words—and the Abstraction (AB)—the abstract thought underpinning the text, as estimated by the use of abstract nouns or words—respectively. Similarly, the Discourse Attribute Analysis Program (DAAP [75]) is based on dictionaries and measures the text producer’s mental activity of connecting affective and cognitive domains. Furthermore, the Linguistic Inquiry and Word Count (LIWC [76]) detects specific keywords assumed as the marker of relevant characteristics of cognitive processing (e.g., emotions, cognitions, or perceptions). However, in spite of their validity, reliability, and spread, these measures imply an invariant value of the words composing the dictionaries. Thus, they do not take into due account the indexicality of meaning [44], due to the field-dynamic nature of meaning-making.

## 3. Aims and Hypotheses

A preliminary version of the ASI was used in a recent study analyzing the evolution of meaning characterizing the dreams of a patient through the course of psychotherapy [77]. In the context of that study, the ASI proved successful in estimating the saturation of the affect-laden meanings in the patient’s dreams. Analyses showed that the saturation followed a meaningful, though nonlinear, trajectory, globally indicative of the progressive increase in the patient’s capacity to elaborate unconscious, affectively relevant areas of her mental landscape. Such findings can be viewed as encouraging, preliminary, indirect evidence in support of the ASI; however, results are based on a single case study and do not provide information on the relationship between the ASI and other measures of affect.

The present study starts from these preliminary findings and is aimed at providing an initial validation of the ASI, with a specific focus on its validity as a measure of affective intensity. More particularly, it intends to test the two core assumptions underlying the ASI: (i) the ASI is able to detect the affective saturation of a text, and (ii) the affective saturation of the text reflects the intensity of the affective state characterizing the meaning-maker involved in producing the text. This gives rise to the following three hypotheses.

First, an association between the ASI and an independent, content-based textual index of the affective saturation of the text is expected. Given that no other direct measure of a text’s affective saturation was found in the literature, we developed an ad hoc indirect index of this characteristic, not related to the ASI: *semantic complexity*. Semantic complexity is the degree of variability of textual content, namely the heterogeneity of the spectrum of content active within the text—the more heterogeneous it is, the greater its semantic complexity. We assume that semantic complexity is inversely associated with affective saturation: the lower the former, the higher the latter. This assumption derives from the definition of affective saturation; insofar as one assumes that affective saturation consists of the magnitude of the contribution of affective meaning compared to other meanings, then the greater the affective saturation, the less the contribution provided by other semantic components, and therefore the lower the global variability of the text. Two studies provide support for the interpretation of semantic complexity as a proxy of affective saturation. First, in the context of the analysis of a therapist–patient exchange, the semantic complexity of narratives proved to be associated inversely with an index of the relevance of generalized affective meanings [78]. Second, in the context of a study of European societies’ cultural milieus, secondary quali-quantitative analyses highlighted that the semantic richness of the cultural worldviews identified proved to be inversely associated with their affective saturation [20] (see above paragraph, *A geometric model of affective saturation*). An analysis of the association between the ASI and semantic complexity was carried out, paying attention to checking the potential effect of the lengths of the texts under investigation (estimated in terms of the number of words).

Second, the ASI is expected to be associated with an independent measure of the intensity of the meaning-maker’s affective state related to text production. To this end, a physiological index of propensity to engage in context-appropriate affective responses or affective arousal was adopted: heart rate variability (HRV), a measure of parasympathetic autonomic nervous system function. The relationship between the ASI and HRV was estimated by parsing out the potential effect of the individual’s capacity for affective regulation. This was done because it is plausible to think that the impact of the intensity of the affective state on the meaning-making underpinning the production of the text, namely the affective saturation, is moderated by the meaning-maker’s capacity to “filter” her/his affective activation. One can expect that the lower the capacity for affective regulation, the weaker the elaborative filter, and, therefore, the stronger the relationship between affective intensity and saturation.

Third, the ASI is expected to be associated with an independent, content-unrelated impact of the affective intensity on the text. To this end, we implemented lexical-syntactic complexity as a proxy of that impact. This choice is based on the combination of the following ideas. First, the lexical-syntactic complexity reflects the efficiency of the meaning-making underpinning the text production [79,80]. The production of lexically and syntactically complex texts (e.g., texts comprising long sentences, based on networks of multi-level hierarchized statements) requires computational efficiency (e.g., working memory and abstract conceptualization); correspondingly, low-efficiency meaning-making reduces the text’s lexical-syntactic complexity. Second, affective intensity prevents the computational and functional efficiency of meaning-making—e.g., it reduces metacognitive processes, the availability of working memory, and access to abstract reasoning [4,81,82]. Thus, affective intensity has a negative impact on syntactic complexity, via the reduction in the efficiency of the underpinning meaning-making. There is already indirect empirical evidence of the inversely proportional association between lexical-syntactic complexity and affective mental state. (A) It has been shown that in patients with schizophrenia—i.e., patients who are characterized by overwhelming affective states—the number of words per sentence in spontaneous speech is significantly lower than in patients with less severe psychiatric diseases and in non-clinical samples. It is worth adding that this is not due to an impairment in the global production of narratives, as shown by the fact that individuals with schizophrenia produce narratives with approximately the same number of words as control groups do [83]. (B) DePaulo and colleagues [84] reviewed empirical studies of deception cues. They moved from the well-supported premise that lie-telling is more emotionally challenging than telling the truth [85]. On this basis, they highlighted that research converges on the finding that the self-presentations of liars, in their free verbal narratives, are characterized by shorter responses and simpler syntactical configurations. (C) In the context of the Terror Management Theory [86], it was found that individuals who received prime activating meanings related to one’s death—assumed to trigger deep states of anxiety—generated shorter (in terms of fewer words and fewer letters per word) autobiographical narratives compared to controls. The analysis of the relationship between the ASI and syntactic complexity was carried out while paying attention to controlling the level of affective regulation. This control is expected to address potential bias due to the fact that the affective saturation measured by the ASI and syntactic complexity can be influenced by differing abilities to regulate affect. In summary, the alternative hypotheses tested against the null hypotheses were: 

**Hypothesis** **1** **(H1)**.*ASI is negatively associated with text semantic complexity*.

**Hypothesis** **2** **(H2)**.*ASI is positively associated with affective intensity*.

**Hypothesis** **3** **(H3)**.*ASI is negatively associated with lexical-syntactic complexity*.

The first hypotheses concern the ASI’s convergent validity; the others concern concurrent validity.

## 4. Method

### 4.1. Sample

The study used a convenience sample of 42 academic students with Italian as their mother tongue. Participants were excluded in the case of self-reported (current or past) psychiatric diagnoses or if they reported psychopathological symptoms over the threshold of clinical relevance (the SCL-R’s GSI index was adopted to this end, see below Section 4.3.1). As a result, 2 participants were not included in the analysis. Thus, the sample consisted of N = 40 (34 F; age: M = 25.33; SD = 2.77).

### 4.2. Procedure

Participants, screened as to the exclusion criterion (presence of current or past psychiatric diagnoses), were contacted through a snowballing procedure. Each participant was invited to the laboratory, provided with a description of the study, informed about the procedure, and asked to sign a written informed consent (Ethical Committee of the Department of Dynamic and Clinical Psychology and Health studies, Sapienza, University of Rome; Prot. n. 0000453). Then, ECG electrodes were attached, and a 5 min assessment at rest was obtained. After that, the participant was asked to fill out the questionnaire (SCL-90-R and DERS) and to undergo the semi-structured interview.

The semi-structured interview was carried out by a trained clinical psychologist, unaware of the aims of the study. The interview focused on neutral issues concerning participants’ involvement in their academic course as well as on life and lifestyle matters. It ended after 10 min. In Table 1, some examples of the questions asked to participants are reported. Interviews were audio-recorded and then subjected to verbatim transcription, from which the textual-based indexes (ASI, SEI, and FVI) were derived.

The choice of basing the interview on neutral issues responded to the aim of focusing on the participants’ baseline levels of affective activation. This was done with the twofold purpose of controlling for potential bias due to the between-subject variability in ways of coping with higher levels of affective activation and of avoiding a potential ceiling effect, reducing data variance.

### 4.3. Measures

The study implemented the following measures:(a)Symptom CheckList-90-Revised (SCL-90-R; [87]), to assess the presence of psychopathological symptoms, considered the exclusion criterion.(b)Difficulties in Emotion Regulation Scale (DERS; [88]), used to measure the participant’s capability for affective regulation.(c)Affective Saturation Index (ASI).(d)Semantic Entropy Index (SEI), used to measure the text’s semantic complexity.(e)Flesch–Vacca Readability Index (FVI; [89]), used to measure the text’s syntactic complexity.(f)Resting Heart Rate Variability, (indexed by the root mean square of the successive differences between normal heartbeats; rMSSD [90]) an index of parasympathetic control of the heart, used to measure the participant’s propensity for affective arousal.

#### 4.3.1. Symptom Check List 90-Revised (SCL-90-R)

The SCL-90-R [87] is a widely used self-report multidimensional inventory, measuring a broad range of symptoms. The SCL-90 has been validated over various populations [91,92,93]. The SCL-90 measures the symptomatic intensity of mental and physical impairment over the last 7 days. It consists of 90 items answered on a five-point Likert scale with a score of 0 (not at all) to 4 (extremely). The inventory fits a 9-factor structure (Cronbach’s alphas ranging from 0.77 for psychoticism to 0.90 for depression) and provides a Global Severity Index (GSI), which is considered a reliable indicator of the current level of overall psychological distress [94,95]. In the present study, we adopted a GSI of <60 as the exclusion criterion. This threshold is, in fact, indicative of clinically relevant conditions of psychological distress [93,94,95,96].

#### 4.3.2. Difficulties in Emotion Regulation Scale (DERS)

The DERS [88] is a 36-item self-report measure developed to assess multiple facets of emotion regulation, including abilities to identify, differentiate, and accept emotional experiences, engage in goal-directed behavior, inhibit impulsive behavior in the context of negative emotions, and use effective emotion modulation strategies.

The DERS’ items are rated on a five-point Likert scale ranging from 1 (almost never) to 5 (almost always). Despite the fact that the original 6-factor structure is debated [97,98,99,100,101], there is agreement on the DERS total score as an index of emotional impairment or dysregulation (Cronbach’s alpha = 0.93) [89,101,102,103,104,105]. Higher scores in the DERS indicate more difficulties in emotion regulation [106]. The DERS is proven to have high internal consistency and test–retest reliability and good predictive and construct validity [103]. The DERS is proven to be sensitive to change due to successful clinical intervention [106,107] and to be correlated with behavioral measures of emotion dysregulation [103]. Based on these results, the measure has gained wide acceptance as a reliable and valid measure of emotion dysregulation in adults [107].

In the present study, we adopted the total DERS scores as an independent measure of the individual’s capability for affective regulation, introduced to control the effect of this facet on the relationship between the ASI and independent measures of affective activation and as well as text syntactic complexity.

#### 4.3.3. Affective Saturation Index (ASI)

The ASI estimates affective saturation by computing the contribution of the primary dimensions of the semiotic space, which models the meaning of the textual corpus under analysis (see above, Section 2.2). The ASI does this through the automated procedure of textual analysis (ACASM [108,109]) described below.

##### ACASM Procedure

ACASM is a computer-based procedure of textual analysis. ACASM can be implemented by several kinds of software. The current analysis used T-Lab [110,111]. T-Lab is appliable to any text based on the Latin alphabet. For several languages (e.g., English, Italian, French, Spanish, German) T-Lab is able to perform both text preprocessing (e.g., the disambiguation of words) and all computations in a fully automatized way. However, these computations are inspectable, their parameters (e.g., the number and type of words under analysis) can be modulated, and their outputs adjusted by the researcher in view of the specific aim of the investigation.

Below, the main steps of the ACASM procedure are described, as it is implemented by the T-Lab software.

(A) The text is automatically segmented into Elementary Context Units (ECUs) according to the following criteria: (i) each ECU begins just after the end of the previous ECU; (ii) each ECU ends with the first punctuation mark (‘‘.’’ or ‘‘!’’ or ‘‘?’’) occurring after a threshold of 250 characters from the first character; and (iii) if an ECU is longer than 500 characters, it ends with the last word within this length, even if there is no punctuation mark.

(B) The software builds a list of all lexical forms (i.e., any string of characters comprised between two blank spaces) present in the transcripts.

(C) The lexical forms are subjected to lemmatization, namely any lexical form is tracked back to the lemmas it belongs to (e.g., lexical forms “I go,” “you went,” and “they are going” are classified as the lemma “to go”). For many languages, T-lab can perform this step in a fully automatized way.

(D) Instrumental lexical forms (e.g., “to,” “and,” “of,” etc.), as well as lexical forms devoid of meaning (e.g., due to typos) are excluded. This operation, too, is carried out automatically.

(E) ACASM procedures set T-Lab to exclude from the analysis the first 5% of the most frequent lemmas. This is because very high-frequency lemmas tend to co-occur in too many different ECUs, thus reducing their ability to discriminate among different patterns of co-occurrence. Then, only 10% of the most frequent lemmas of the remaining list are selected. This choice is aimed at reducing the dispersion of information within the dataset. In sum, the analysis is limited to lemmas comprised between the fifth and fifteenth percentile of frequency distribution.

(F) A digital matrix of the text is generated, with the segments of text produced in step A (i.e., the ECUs) as rows and the selected lemmas as columns. The ij-th cell assumes a value of 1 if the j-th lemma is contained in the i-th ECU; a value of 0 is assigned otherwise. Appendix A reports an example of such a digital matrix.

(G) The ECU ∗ lemma matrix is subjected to a Lexical Correspondence Analysis (LCoA). The output of the LCoA is a factorial space, each factorial dimension of which maps a component of the meaning active within the whole textual corpus (in the current analysis, the whole set of interviews). Moreover, the LCoA output provides the coordinate on the factorial space of each ECU, as well as of any super-ordered categories in terms of which the ECUs are classified (in the current analysis, the interview). Henceforth, the object of the factorial coordinate (i.e., ECU or super-ordered category) is denoted as a text unit. The factorial coordinate is the measure of the degree of association between the text unit and the factorial dimension—the higher the coordinate, the higher the association, therefore the higher the contribution of that factorial dimension to the meaning of that text unit. In the current study, we focus on the factorial coordinates of the interviews as text units. However, depending on the researcher’s aim, the ASI can also be computed by taking the ECU as a text unit. In doing so, a more fine-grained map of the ASI trends throughout the text would be obtained, with a value of ASI for each ECU—i.e., for each sentence. For instance, take the case of a text comprising the verbatim transcript of a whole psychotherapy course. The ASI can be computed either by adopting each session as a text unit (i.e., at the level of super-ordered category) or at the level of a single ECU as a text unit. In the former case, the ASI trend would concern the whole psychotherapy course; in the latter case, the focus would be on the moment-by-moment within-session ASI evolution. Incidentally, the fine-grained focus could be used to analyze the interplay between the participants of the exchange (note that the latter kind of analysis requires the researcher’s intervention to differentiate the ASI trends of the interlocutors).

(H) Factorial coordinates are used to compute the ASI. More specifically, the ASI is calculated for each unit (in the current analysis, each interview) as its Euclidean distance from the origin of the factorial space. It is worth highlighting that the Euclidean distance is computed by using the coordinates of the first two factorial dimensions only, as in Equation (1). This is so because the ASI assumes that the first two dimensions are the computational equivalent of the PSM primary dimensions (see Section 2.2):(1)$${\mathrm{ASI}}_{t}=\sqrt{C_{F1\left(t\right)}{}^{2}+C_{F2\left(t\right)}{}^{2}}$$
where ASI_t_ is the Affective Saturation Index of the textual unit *t* and C_F1(*t*)_ and C_F2(*t*)_ are the factorial coordinates of the textual unit *t* on the first and second factorial dimension, respectively.

From Equation (1), it can be seen that the ASI increases when one or both of the factorial coordinates increase. Accordingly, the ASI can be interpreted as a measure of the magnitude of the contribution of the two first dimensions of the factorial space to the meaning of the textual unit—the higher the ASI, the greater the contribution. Therefore, the ASI can be considered an index of the degree of saturation of the affective meanings comprising the textual unit.

#### 4.3.4. Semantic Entropy Index (SEI)

To estimate the semantic complexity, we used an ad hoc index: the *Semantic Entropy Index* (*SEI*). The SEI is derived from the Information Entropy formula [112], applied to the text’s thematic contents. However, previous studies which applied Informative Entropy to text analysis focused on lexical features [113], language comprehension, meaning representation [114,115], and associative strengths among words [111,115]. Different from these studies, the SEI is aimed at detecting the amount of semantic variability in the text. The higher the SEI, the higher the content variability (i.e., the complexity) of the text under analysis is. In the current study, the SEI was applied to each interview.

The SEI was estimated in accordance with the following procedure. First, the digital matrix of the textual corpus (i.e., the same matrix used in the first stage of the computation of the ASI, i.e., the LCoA) was subjected to a Lexical Cluster Analysis (LClA). The LClA groups sentences (Elementary Context Units, ECUs) that tend to share the same co-occurring lemmas. In this way, each cluster can be considered indicative of a thematic content active in the textual corpus and characterized semantically by the pattern of co-occurring lemmas making those ECUs similar to each other (for details, see [108,109]). The number of clusters in which the text is segmented is established in accordance with an iterative algorithm [110,111]; the procedure of clustering stops when further partitions produce no significant improvement of the inter-/intra-cluster ratio, which means that increasing the number of clusters does not produce an appreciable increment of information. In the current analysis, the LClA generated 5 clusters/thematic contents as optimal partitions. Accordingly, each interview was characterized by the relative frequency of each cluster/thematic content within it—i.e., the proportion of ECUs covered by each cluster within the interview.

The SEI was computed in accordance with the following formula:(2)$$\mathrm{SEI}=-\overset{n}{\underset{i=1}{\sum }}\;p\left(x_{i}\right)\;ln\;p\left(x_{i}\right)\;\;$$
where SEI is the degree of the Semantic Entropy Index of the interview *I*, *n* is the number of clusters (i.e., thematic contents) obtained by the LClA, and p(*x_i_*) stands for the probability that a cluster *x_i_* occurs.

According to Equation (2), the entropy consists of the homogeneity of the probability of the occurrence of clusters/thematic content. The more each cluster has an occurrence probability similar to that of others, the more the thematic variability of the text, thus its complexity. This means that, in the context of the current analysis, the highest SEI is given if each cluster/thematic content has f = 0.2 (i.e., 1/5) of relative frequency. In that case, indeed, the text could be characterized by one or more specific clusters; rather, all clusters need to be considered to represent its content.

#### 4.3.5. Flesch–Vacca Index (FVI)

The FVI is a measure of text readability [89]. Text readability is the extent to which a text can be understood by a reader as a result of its lexical and syntactic characteristics. Specifically, the FVI is an Italian language adjustment of the Flesch [116,117] Reading Ease formula. The FVI assumes that long words are typically used less frequently than short ones and that long sentences are usually more complex, from a syntactical point of view, than short ones.

The FVI was computed for each interview separately, in accordance with the following formula (see Equation (3)):FVI_t_ = 217 − (1.3 ∗ ASL_t_) − 0.6 ∗ AWL_t_(3)
where ASL_t_ is the length of sentences of the text unit *t*, measured as the average number of syllables in 100 words, and AWL_t_ denotes the length of the sentences, expressed as the average mean of words per sentence.

#### 4.3.6. Heart Rate Variability (HRV)

HRV is the variability of the time between adjacent heartbeats, resulting from the dynamic interplay between the fast-acting parasympathetic nervous system and the relatively slower sympathetic nervous system [118]. A higher resting-state HRV reflects better adaptive and flexible prefrontal inhibition to meet various situational demands [119]. In contrast, a lower resting HRV has been related to hypoactive prefrontal regulation, leading to hyperactive subcortical structures and the release of physiological defensive responses [120,121]. Tonic HRV was assessed through the following procedure. A 5 min baseline was started, during which time the participants browsed a gardening magazine (i.e., “vanilla baseline” [122,123,124]), while their beat-to-beat intervals were recorded using the Bodyguard 2 (Firstbeat). The device has been shown to provide reliable measures of beat-to-beat intervals [124].

HRV analyses were performed using Kubios HRV software [125]. This software uses an advanced detrending method based on smoothness priors formulation in which the filtering effect is attenuated at the beginning and the end of the data, thus avoiding the distortion of data end-points. Moreover, the frequency response of the method is adjusted with a single smoothing parameter, selected in such a way that the spectral components of interest are not significantly affected by the detrending [126]. Kubios HRV includes two methods for correcting any artifacts and ectopic beats: (1) a threshold-based correction, in which these are simply corrected by comparing every RR interval value against a local average interval; and (2) automatic correction, in which artifacts are detected from a time series consisting of differences between successive RR intervals.

HRV was mapped by means of the rMSSD, which reflects the vagal regulation of HR [126]. A higher resting rMSSD reflects better psychological and emotional flexibility and the capability to engage in context-appropriate responses [127,128,129]. Accordingly, we used the rMSSD as an index of the individual’s disposition for affective arousal—the lower the rMSSD, the higher the propensity to be subjected to affective arousal (for a similar interpretation of the index, see [118,130,131]). As the distribution of the rMSSD was non-normal, the variable was transformed into its natural logarithms.

### 4.4. Data Analysis

A set of one-way ANOVAs and bivariate correlations were first used to estimate the size of the relationships among indexes and sample characteristics (age, gender) and between them. Then, three multiple linear regressions (standard multiple regression method) were performed, one for each hypothesis:

H1 was tested by means of a regression model, with the index of saturation (SEI) as the dependent variable and the ASI and the length of the interview (measured by the number of words) as predictors. The latter was introduced in order to control for its effect on the SEI–ASI relationship.H2 was tested by means of a regression model, with the physiological measure (rMSSD) as the dependent variable and the ASI as the predictor. The DERS was introduced in the model as a further predictor, in order to estimate the rMSSD–ASI association and the net effect of the individual’s capacity for emotion regulation.H3 was tested by means of a regression model, with the index of syntactic complexity (FVI) as the dependent variable and the ASI as a predictor. We introduced the length of the interview (in number of words) and the DERS as further predictors, to control for their effects on the SEI–ASI relationship.

## 5. Results

Table 2 reports descriptive statistics of the variables adopted. As shown by the values of kurtosis and skewness, the indexes (in the case of the rMSSD, after logarithmic transformation) proved to approximate the normal distribution (the distribution of ASI is shown in Figure 1).

No significant differences resulted between males and females as to the level of the ASI (ANOVA test: F [1,38] = 0.337, *p* = 0.565). Accordingly, and considering that the sample is characterized by a higher proportion of women than men, we did not carry out separate analyses. The length of the interview (in number of words) did not correlate with any of the indexes examined. However, we have used this as a control variable in regression models 1 and 3, given that these models have a text-based index as the dependent variable.

Table 3 reports the correlations between the main variables of the study. The ASI proved to be negatively associated with the SEI (r = −0.657), rMSSD (r = −0.468), and FVI (r = −0.426). No significant correlations emerged for Age, Words, DERS, and SCL-90R. The log-transformed rMSSD and the FVI proved to correlate robustly (r = 0.489).

All three multiple regression models proved to be significant (*p* < 0.001, *p* < 0.004, and *p* < 0.005, respectively; cf. Table 4). Table 5 and Table 6 report the main parameters of regression model 1, with SEI as the dependent variable and the ASI and Words (i.e., the number of words in the interview) as predictors. The model did not suffer from problems of collinearity (VIF = 1.007); the adjusted R square was 0.428 (std. err. of estimation = 0.101). The inclusion of Words did not modify the parameters of the model significantly (change of R from model 1 and model 2: *p* = 0.199; cf. Table 5). The ASI beta coefficient was −0.644 (t = −5.296, *p* < 0.000); the Words beta coefficient (0.159) was not significant (cf. Table 6). The distribution of residuals approximated the normal distribution (Figure 2).

Table 7 and Table 8 report the main parameters of regression model 2, with the rMSSD as the dependent variable and the ASI and DERS as the predictors. The model did not suffer from problems of collinearity (VIF = 1.001); the adjusted R square was 0.214 (std. err. of estimation = 0.549). The inclusion of the DERS did not modify the parameters of the model significantly (change of R from model 1 and model 2: *p*= 0.192; cf. Table 7). The ASI beta coefficient was −0.472 (t = −3.325; *p* < 0.002); the DERS beta coefficient was not significant (0.189; cf. Table 8). The distribution of residuals approximates the normal distribution (Figure 3).

Table 9 and Table 10 report the main parameters of regression model 3, with the FVI as the dependent variable and the ASI, DERS, and age as the predictors (we included age as a covariate, due to its high correlation with the FVI; this means that in this case the model was calculated on n = 39 group, given that 1 participant had unknown age). The model did not suffer from problems of collinearity (VIFs close to 1); the adjusted R square was 0.241 (std. err. of estimation = 5.409). The inclusion of the DERS and age did not modify the parameters of the model significantly (*p* = 0.064; cf. Table 9). The ASI beta coefficient is −0.301 (t = −2.187; *p* < 0.035); the age beta coefficient is also significant (−0.364, *p* < 0.002). The DERS beta coefficient (.029) was not significant; (cf. Table 10). The distribution of residuals approximates the normal distribution (Figure 4).

## 6. Discussion and Conclusions

As hypothesized, the ASI proved to be significantly and inversely associated with the SEI, the independent proxy of affective saturation. The association was robust, in the expected direction (ASI beta = −0.644), and was not weakened by the control of the potential effect of the length of the interviews.

Second, the ASI was significantly and inversely correlated with the rMSSD, the physiological index of the disposition for affective arousal. This relationship was in the expected direction (once one considers that a lower rMSSD indicates higher arousal), and robust, both when estimated directly (r = −0.468) and once the index of affective regulation (DERS) was introduced in the regression model (ASI beta = −0.472). This finding suggests that the ASI is able to detect the intensity of the affective state of the meaning-making involved in the production of the text, in a way that is not influenced by the individual’s capability to regulate affective arousal.

Finally, these findings are consistent with the idea that the ASI is able to detect the lexical-syntactic complexity of the text, assumed as an independent correlate of the affective intensity. As to this latter assumption, it is worth highlighting that it was supported by the high correlation between the index of lexical-syntactic complexity (FVI) and the rMSSD (r = 0.489). These results are consistent with and further support lines of thinking that view lexical-syntactic complexity as a property of the textual output that is subject to the influence of the affective state over the meaning-making underpinning its production. The ASI proved to be associated with the FVI (ASI beta = −0.323), in the expected direction. Moreover, in this case, the effect was estimated after the individual’s capability for affective regulation was checked.

Taken as a whole, the present findings support both the aspects of the ASI construct and the criterion validity investigated. The ASI proved to be a measure capable of detecting the structural organization of textual meaning—more specifically, of estimating to what extent the text is saturated by affective meaning. Moreover, the ASI’s measurement of affective saturation proved to be a valid estimator of the producer’s physiological affective state at rest as well as of the impact of affective intensity on meaning-making (as marked by the text’s syntactic complexity).

These findings are promising for their theoretical, methodological, and practical implications. From a theoretical standpoint, they enforce the semiotic framework upon which the ASI is based. What needs to be highlighted here is that the intensity of the meaning-maker’s affective state proved to be associated not only with the efficiency of the cognitive process underpinning the text production—as signaled by the relationship between affective intensity and lexical-syntactic complexity—but also with the inherent structural organization of the textual meaning—i.e., the relevance of the primary dimensions of meaning over the others, which is the specific property on which the ASI focuses. This legitimates the semiotic view of affect—namely, the idea that affect is an embodied form of meaning that, due to its nature, operates directly on the text’s semantic organization. In other words, the findings are consistent with the ASI viewpoint, which does not see affect as an exogenous factor influencing the text from the outside. Rather, it conceives affect as an inherent characteristic of the text; affect does not work on meaning-making by constraining or channeling it but is part and parcel of it [20]. From a methodological standpoint, the ASI opens new opportunities for measuring affect. Two characteristics of the ASI are worth mentioning here. First, the ASI is almost completely insensitive to the size of the data—this means that it can be implemented on a large textual dataset, therefore enabling large-scale studies that can link individual and social levels of analyses as well as studies based on the density of units of observation required for dynamic time series. Second, the affective meaning is frozen, as it were, in the text. Therefore, the ASI’s use of texts as a source of the measurement of affective intensity enables the off-line detection of that dimension, namely the possibility for measuring affect in an independent spatio-temporal setting with respect to the setting in which it was activated. The combination of these characteristics envisages thrilling new opportunities—e.g., large-scale retrospective time-series analyses, to model the dynamics of affective activation characterizing the socio-cultural historical evolution of given social groups, and time-series analyses to map the dynamic evolution of meaning-making over the psychotherapy process [77]. The ASI’s methodological flexibility has practical implications, too. One can envisage a plurality of applications of the method, in the many fields where the measurement of affect and its impact on meaning-making can be relevant for both interpretative and interventional aims—e.g., fields such as clinical and community interventions, social communication, marketing, and media monitoring (on the role of affective sensemaking in society [27]).

However, the fact that the findings are encouraging must not lead us to underestimate the limitations of the study. There are three main shortcomings to highlight. First, the study was based on a convenience sample of Italian students in a degree course in psychology characterized by homogeneous age and a higher proportion of women. This made it impossible to test the role of language, age, and gender on the relationship between the ASI and other indexes and therefore generalize the findings beyond this specific group.

Second, the study adopted a psychophysiological index (rMSSD) assessed at rest, which is a trait measure. It was implemented before the interview to estimate the participants’ baseline disposition to a given level of affective intensity and capacity for affective regulation, respectively. Thus, it is not a direct measure of the level of affective intensity— and its variation—during the interview. We did this because the adoption of a state measure, mapping the ongoing physiological state of the participants alongside the interview, would have involved a level of computational complexity (e.g., the necessity to match the ASI values that took the whole interview as a unit of analysis and the instant-by-instant physiological values) which would have been outside the scope of the current study, which is aimed at the first stage of the validation of the ASI. Furthermore, while it must be recognized that the HRV index only partially captures the affective state occurring during the interview, it must also be highlighted that the use of this index is a conservative choice, which underestimates the relationship between the ASI and the participants’ affective intensity manifested during the interview. This is because it is plausible to think that, in response to the interview, individuals would differ in how much their affective state varies from the baseline, as a result of individual differences in personality and other psychological characteristics. Thus, insofar as the ASI is a measure of the current affective intensity on the text, the use of a trait measure weakens the chance to detect the capacity of the ASI to estimate the physiological state underpinning text production.

The third hypothesis was based on an indirect proxy of the impact of affective intensity on meaning-making—the text’s lexical-syntactic complexity. Thus, hypothesis 3 of the study—i.e., the fact that the ASI is able to detect the impact of affective intensity on meaning-making—must be considered only indirectly tested, insofar as one accepts the assumption that lexical-syntactic complexity is a valid marker of the efficiency of meaning-making. This assumption is fostered by the findings of the study (i.e., the high correlation between the FVI and the rMSSD), but is not systematically supported by the literature, which lacks specific studies on this issue.

These issues need to be addressed by the next steps of the ASI’s validation. Further studies will be implemented to test the ASI’s validity on other groups (e.g., lower educated people and clinical populations) and other kinds of texts (e.g., highly affect-laden communications and texts characterized by positive vs. negative affective valence). Finally, deeper analyses of the specific mechanisms underpinning the relationship between the textual and physiological components of affect, as well as the role played by the regulative cognitive processes in that relationship, will be brought into focus.

## Figures and Tables

**Figure 1 entropy-23-01421-f001:**
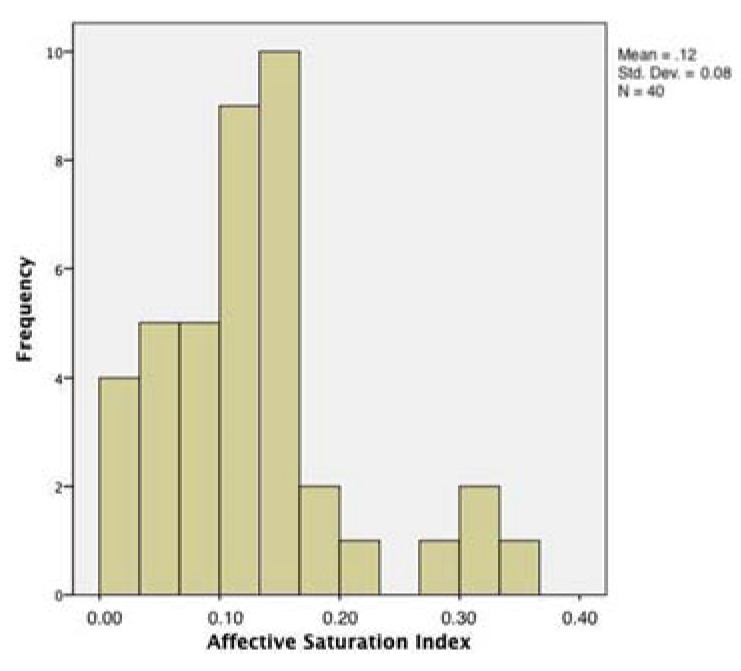
Affective Saturation Index (ASI) distribution in the sample under analysis.

**Figure 2 entropy-23-01421-f002:**
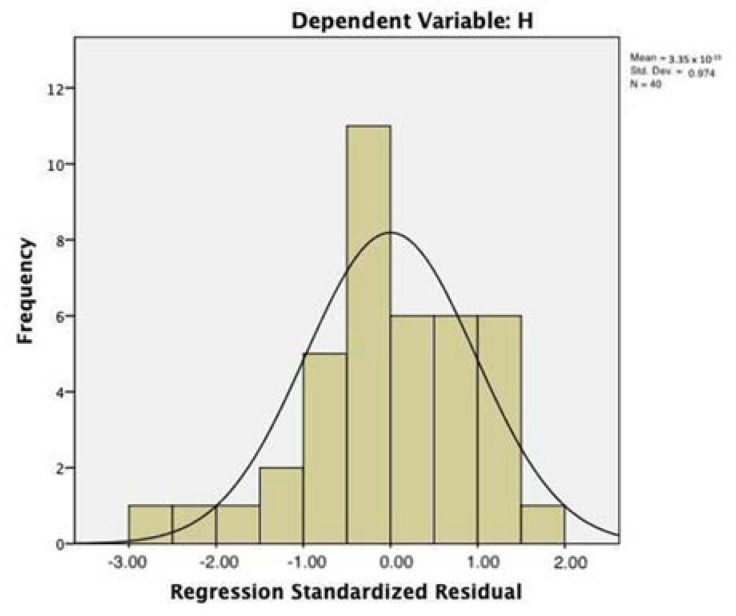
Regression model with the index of saturation (SEI) as the dependent variable. Distribution of residuals.

**Figure 3 entropy-23-01421-f003:**
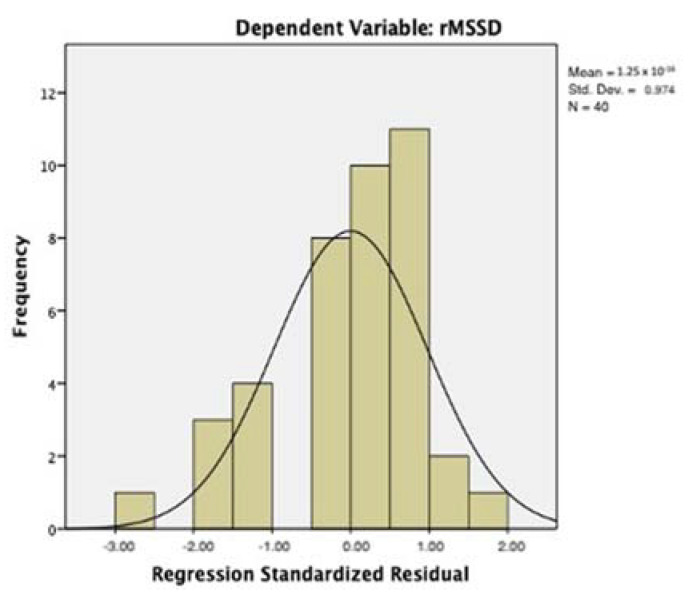
Regression model with the log-transformed root mean square of the successive differences between normal heartbeats (rMSSD) as the dependent variable. Distribution of residuals.

**Figure 4 entropy-23-01421-f004:**
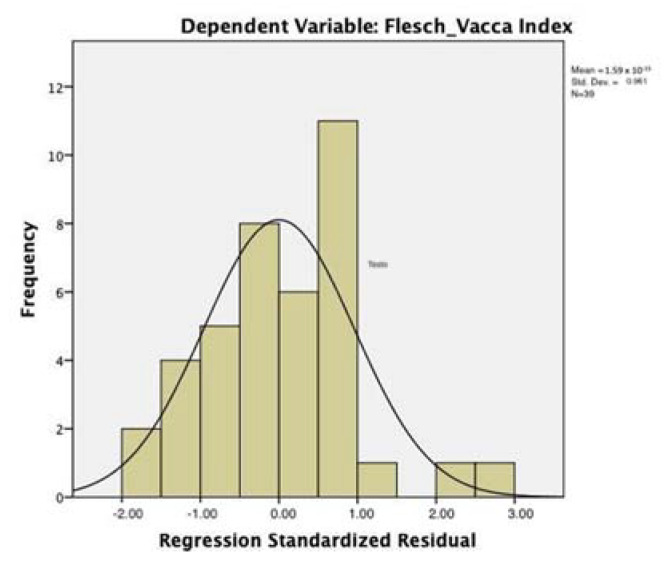
Regression model with the FVI as the dependent variable. Distribution of residuals.

**Table 1 entropy-23-01421-t001:** Examples of questions used in the semi-structured interview.

Which degree program are you enrolled in?
What is the subject that you are most passionate about?
Whom do you live with?
Are you still with your family or do you live alone?
Do you have any brothers or sisters?
What do you like to do together?
What do you like to do in your free time?
Do you have any friends?
Do you have any hobbies?

**Table 2 entropy-23-01421-t002:** Descriptive statistics of the measures under analysis.

	N	Mean	Min	Max	Std. Dev.	Skewness	SE	Kurtosis	SE
Age	39	25.107	21	31	2.80784	0.348	0.378	−0.879	0.741
SCL90-R	40	44.025	33	75	11.226	1.252	0.374	0.498	0.733
ASI	40	0.124	0.01	0.34	0.08	1.134	0.374	1.391	0.733
SEI	40	1.456	1.081	1.592	0.133	−1.358	0.374	1.091	0.733
rMSSD	40	3.846	2.25	4.94	0.62	−0.885	0.374	0.242	0.733
FVI	40	70.651	54	80	6.133	−0.79	0.374	0.683	0.733
Words	40	1362.43	988	1746	170.569	0.035	0.374	−0.021	0.733
DERS	40	85.75	49	128	21.584	0.367	0.374	−0.651	0.733

*Note.* Age N = 39 given that 1 participant had unknown age; SCL90-R = Symptom Checklist-90 Revised; ASI = Affective Saturation Index; SEI = index of saturation; rMSSD = log-transformed root mean square of the successive differences between normal heartbeats; FVI = index of syntactic complexity; DERS = Difficulties in Emotion Regulation Scale.

**Table 3 entropy-23-01421-t003:** Pearson correlations between the measures under analysis.

	SEI	rMSSD	FVI	Age	Words	DERS	SCL-90R
ASI	−0.657 *	−0.468 *	−0.426 *	0.288	−0.084	0.023	0.047
SEI		0.464 *	0.388 *	−0.247	0.213	−0.012	−0.053
rMSSD			0.489 *	−0.064	−0.012	0.178	0.156
FVI				−0.452 *	0.026	−0.034	−0.59
Age					−0.212	0.155	0.147
Words						0.133	0.084
DERS							0.653 *

* significant correlations at 0.01 level two-tailed. ASI = Affective Saturation Index; SEI = index of saturation; rMSSD = log-transformed root mean square of the successive differences between normal heartbeats; FVI = index of syntactic complexity; DERS = Difficulties in Emotion Regulation Scale; SCL-90 = Symptom Check List-90 Revised.

**Table 4 entropy-23-01421-t004:** Regression models. ANOVA test.

	Sum of Squares	df	Mean Square	F	Sig.
*Regression Model 1: Dependent Variable: SEI; Predictors: ASI, Words*					
Regression	0.315	2	0.158	15.58	0.001
Residual	0.375	37	0.010		
Total	0.691	39			
*Regression Model 2: Dependent Variable: rMSSD; Predictors: ASI, DERS*					
Regression	3.813	2	1.907	6.313	0.004
Residual	11.175	37	0.302		
Total	14.988	39			
*Regression Model 3: Dependent Variable: FVI; Predictors: ASI, DERS, age*					
Regression	441.309	3	147.103	5.028	0.005
Residual	1023.922	35	29.255		
Total	1465.231	38			

*Note.* ASI = Affective Saturation Index; SEI = index of saturation; rMSSD = log-transformed root mean square of the successive differences between normal heartbeats; FVI = index of syntactic complexity.

**Table 5 entropy-23-01421-t005:** Regression model with the index of saturation (SEI) as the dependent variable. Summary of the model.

Model *	R	R^2^	Adjusted R^2^	Std. Error	F Change	df1	df2	Sig. F Change
1	0.657	0.432	0.417	0.101				
2	0.676	0.457	0.428	0.101	1.714	1	37	0.199

* Model 1 predictor: Affective Saturation Index (ASI); Model 2 predictors: ASI, Words.

**Table 6 entropy-23-01421-t006:** Regression model with the index of saturation (SEI) as the dependent variable (model 2, with all predictors included).

	B	Stand. Error	Beta	t	Sig.	VIF
Constant	1.42	0.135		10.55		
ASI	−1.064	0.201	−0.64	−5.3	0	1.007
Words	0	0	0.159	1.309	0.199	1.007

*Note.* ASI = Affective Saturation Index.

**Table 7 entropy-23-01421-t007:** Regression model with the log-transformed root mean square of the successive differences between normal heartbeats (rMSSD) as the dependent variable. Summary of the model.

Model *	R	R^2^	Adjusted R^2^	Std. Error	F Change	df1	df2	Sig. F Change
1	0.468	0.219	0.198	0.55506				
2	0.504	0.254	0.214	0.54956	1.764	1	37	0.192

* Model 1 predictor: Affective Saturation Index (ASI); Model 2 predictors: ASI, Difficulties in Emotion Regulation Scale (DERS).

**Table 8 entropy-23-01421-t008:** Regression model with the log-transformed root mean square of the successive differences between normal heartbeats (rMSSD) as the dependent variable (model 2, with all predictors included).

	B	Stand. Error	Beta	t	Sig.	VIF
Constant	3.835	0.382		10.03	0.001	
ASI	−3.636	1.093	−0.47	−3.33	0.002	1.001
DERS	0.005	0.004	0.189	1.328	0.192	1.001

*Note.* ASI = Affective Saturation Index; DERS = Difficulties in Emotion Regulation Scale.

**Table 9 entropy-23-01421-t009:** Regression model with the index of syntactic complexity (FVI) as the dependent variable. Summary of the model.

Model *	R	R^2^	Adjusted R^2^	Std. Error	F Change	df1	df2	Sig. F Change
1	0.427	0.182	0.16	5.691				
2	0.549	0.301	0.241	5.409	2.978	2	35	0.064

* Model 1 predictor: Affective Saturation Index (ASI); Model 2 predictors: Affective Saturation Index (ASI), Difficulties in Emotion Regulation Scale (DERS), age.

**Table 10 entropy-23-01421-t010:** Regression model with the index of syntactic complexity (FVI) as the dependent variable (model 2, with all predictors included).

	B	Stand. Error	Beta	t	Sig.	VIF
Constant	93.251	8.251		11.3	0.001	
ASI	−24.958	11.41	−0.32	−2.19	0.035	1.091
DERS	0.008	0.041	0.029	0.204	0.839	1.025
age	−0.804	0.331	−0.36	−2.43	0.02	1.118

*Note.* ASI = Affective Saturation Index; DERS = Difficulties in Emotion Regulation Scale.

## Data Availability

Not applicable.

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
