# Peer review of "Affective Saturation Index: A Lexical Measure of Affect"

_entropy, 2021, doi:10.3390/e23111421_

Round 1
Reviewer 1 Report
The paper presents the Affective Saturation Index (ASI), a textual-based measure of the intensity of emotion present in a text.
There are several English flaws. Focusing only on Introduction, the more often the problem is plural flaws: 'is' where should be 'are'.
Regarding the content, the hypotheses should be stated as null and alternative hypotheses. And their test should be done using them.
The results are, in the most, presented, with no discussion. See Table 2: Kurtosis and other values are presented, but not discussed.
The same is done in the Conclusion: the values are repeated, with few evaluations and discussions.
Reviewer 2 Report
Review of the manuscript „Affective Saturation Index. A lexical measure of affects“
The paper introduce the Affective Saturation Index measuring the level of affective intensity of a written text. The method seems to be applicable on various kinds of texts, e.g. interviews or journal articles, and is very interesting for the research on affects. Furthermore, the paper reports analyses regarding the construct and criterion validity based on a sample of N = 40 Italian mother-tongue academic students. Correlation and regression analysis suggest both construct and criterion validity.
The paper seems to be very interesting for the readers of Entropy. However, before publication some points should be revised.
Abstract
- Please include effects size measures in the abstract that describe the strength of associations, e.g. between ASI and HRV.
- “42 individuals completed”: This is misleading. The analyzed sample included only 40 persons.
- “Results support ASI’s construct and criterion validity”: This is an over interpretation of the results. You only studied 40 Italian students! There is no evidence that in other samples (e.g. low education level, or older persons) similar correlation will be found.
- “ASI proved 25 able to detect affective saturation”: In my opinion, this conclusion can not draw from your data because you have no other measure of affective saturation (no “external criterion”).
Section “1. Introduction”
- The paper is submitted in the Journal Entropy (section “Complexity Science in Human Change: Research, Models, Clinical Applications”). Please address the topic of the journal (and special section) more in the introduction and show the relation between the Journal topic and the topic of your paper.
Section “2. The Affective Saturation Index”
- Please include a paragraph that describes the main differences between ASI and other text based measures e.g. SEI or LIWC or Mergenthaler’s approach. On that way, it should be easier for the reader to understand what the ASI measure (and what not). Possibly, there are some methodological strengths of ASI compared to SEI or LIWC? What is the difference between Lexical Correspondence Analysis (LCoA) and Lexical Cluster Analysis (LClA), for example?
- In my opinion, an additional subsection on the measurement would be nice. I recommend the inclusion of appendix A as subsection of section 2. Please highlight that ASI is a computer rating. However, it should be discussed whether the coding is fully automatic. If applicable, please report more details on the preprocessing of interview transcripts. When you apply, for example, the computer program the Linguistic Inquiry and Word Count (LIWC) (Pennebaker, Boyd, Jordan, & Blackburn, 2015), then “eh”, “mh”, and many other things need to done.
- As I understand the ASI methodology right, the ASI score refers “only” to the entire text (e.g. all word spoken by the participant in the interview). In the current research, there is a growing interest on the interplay between both interlocutors (e.g. patient and psychotherapist) (for an application example of LIWC in language style matching in psychotherapy, see Aafjes-van Doorn, Porcerelli, & Müller-Frommeyer, 2020; for an overview on behavioral and physiolocial synchrony in psychotherapy see e.g. Wiltshire, Philipsen, Trasmundi, Jensen, & Steffensen, 2020). Therefore my question: Is it possible to extent the ASI approach so that some kind of matching or synchronization can be measured?
- As I understand your method right, the ASI based on theory’s that map meanings of words into a multidimensional space. I am wondering why only a “total/overall” score is considered. Is it not possible to define sub-scores according to the dimensions of used sematic space? In the LIWC approach, for example, we distinguish between the word category’s function, pronoun, article, etc. We can compute language style matching considering a specific category, e.g. pronouns, or the matching of all categories (the “total” score).
- Please add some information whether the ASI methodology in limited to Italian. What languages can also studied with this approach?
- How valid is your method according to senseless texts, e.g. did you get an ASI value for “fun, fun, fun, fun, blue, blue, loud, loud, loud”?
Section 3. Aims
- P6, L264-269: Please write in more detail what you mean by validity and how (this kind of) validity can be proofed. You say, for example, “ASI is able to detect the affective saturation of a text”. To proof this, you need an external criterion / a “true value” of affective saturation to show that the instrument of interest measure affective saturation as the criterion. To proof the validity of Beck-Depression-Inventory typically diagnostic interviews are used as criterion, to give an example.
Section 4. Methods
- I am sorry to say that, but for a validation study a sample of N = 40 participants is too small. Moreover, your examined only university students (all are psychology students, right?). This means that you examine a specific age group und a specific education level. This should be taken into account when drawing conclusions. You showed only validity for this specific group, not for older persons or persons with lower education or persons with psychological disorder etc. Please extent the sample, e.g. another age group or an additional sample of patients (with the same age). Otherwise the abstract and the discussion section needs to revised in the meaning that validity is only showed for psychology students.
- Another possibility to extent the validation study (and resolve the sample size problem) is an addiational examination of written texts. For example, you can compare emotional neutrals texts (e.g. youtube video how to repair a car) with emotionally texts (e.g. description of how the concert of the favorite band was experienced). In terms of predictive validity the ASI score should be lower in neutral texts than in emotionally texts.
- Subsection 4.3.2, page 8ff.: Based on the current description of method, I do not understand how the method (ASI) work. What, for example, is “the textual context*lexical unit matrix”? You refer to Appendix A. There is no explanation of the matrix for example. Moreover, the description in the appendix is also difficult to understand. What means “lemmatization” or “ECU”? Maybe, the method is easier to understand when you explain the ASI step by step using one or two examples of short texts.
- Page 9, Line 415: The equation suggests that the sematic space/set have two dimensions (F1 and F2) and that a distance is defined in this space/set. Please clarify why the space is limited to two dimensions. In the introduction section you refer to theory’s with five dimensions. What is the meaning of dimension 1 and what is the meaning of dimension 2? Furthermore, I do not understand, do you apply a factor analysis (?) on each text so that you get for each text an individual factorial space? How is the distance defined? For example, what is the distance between “red” and “blue” and is this distance larger than the distance between “joy” and “sad”? To make this clear: For me the method is very interesting. But I am not familiar with “sematic processing”. Therefore, the way from single words to an Euclidian space/set is difficult to understand.
- Page 9, line 434: “digital matrix of the textual corpus”. How is this matrix defined? Could you give some examples? Depends the matrix on the researcher?
- Again, it seems that the Lexical Cluster Analysis works with an Euclidian space so that distance between to points in this space can be computed. How is this possible? Are there pre-defined coordinated for each word?
- Page 9, line 450: What is the thematic content x_i? Please explain the relationship between the thematic content and the latent classes (clusters). As I understand the equation, the SEI measure is some kind of order (more precisely the entropy) of the examined text. Maybe you can explain the methods by two text examples (e.g.: “Blue blue blue” have low entropy (SEI) and “this is interesting” have a larger entropy.)?
- Sub-section 4.3.5: Is ASL_t the word count of analyzed text? If so, then FVI_t and word count should be highly correlated. But they did not (see Tab. 3). Did you compute the FVI_t for multiple parts of the interview and then aggregate all these values to an overall-score for the individual interview?
- Please change the order of the subsections for the used instruments. The DERS can be reported after the SCL90 because both are self-reports. Furthermore, the FVI should be introduced after SEI and ASI because all three instruments measure characteristics of the (interview) text.
- Please report for all self-ratings the reliability (e.g. Cronbachs alpha).
Results
- Table 2 suggests that there is one missing data regarding to the age. Impute this value otherwise all sub-sequent analyses based on N=39 when using age as predictor.
- Table 2: Please report also min and max.
- Table 3: Please report also the correlation with the SCL90
- Table 3: Please include correlations with the gender of participants using point-biseral-correlations.
- Table 3: What correlations are reported Pearson or Spearman? Please add this in the note below the table. Spearman are more appropriate because rMSSD, ASI and word count are not normal distributed.
- Table 4-10: Please reduce the number of tables (ideally, the paper should include not more than five Tables). In my opinion, the hypotheses are proofed already by the correlation analyses. So I am wondering why the regression analyses are conducted. Moreover, the model comparison (studying the effect of word count) is not mentioned in the sub-section “4.4 Data Analysis”. However, I agree that a regression analysis is better than a correlation analysis. I recommend the application of a stepwise backward regression. First, a model is applied including ASI as dependent variable and age, gender (!), SCL90 total score, DERS, rMSSD, FVI, Word count, and SEI as predictors (the so called full model). Then, a stepwise regression is conducted and the final model is reported. It can be assumed that age, gender, and word count will be excluded by the algorithm.
- Table 3-10: The rMSSD was log-transformed (P10, L485). Please include is information also in the notes under the Tables.
- 5, Tab. 6 and Fig. 2: In theses analyses SEI is the dependent variable. Please note that you say in the sub-section Data analysis that ASI is always the dependent variable. I belief the SEI prediction is not necessary.
- P12, L564: Please round “Estimation = .100633” to .101
- P12, L566 “p < .000”: This is not possible. Did you mean “p < .001”? The same holds for some the p-values in Tab. 4 and Tab. 8 and Tab. 10.
Discussion
- You hypothesized that “(HP2) - ASI is positively associated with affective intensity” (P7, L338). However, you found that “Second, ASI was significantly and inversely correlated with rMSSD, the physiological index of disposition to affective activation ... (r = -.468)” (P15, L621). This sounds contradictionary. However, in the methods section you say that lower rMSSD indicate higher arousal (not activation). You should highlight this circumstance in the discussion or you should reword the hypotheses so that the sign of correlation fits better to you hypothesis.
- P15, L 628 “ASI proved able to detect the lexical-syntactic complexity of the text”: Despite the significant correlation, I would not draw this conclusion. In fact, affective saturation and entropy (higher disorder/ higher randomness regarding the used words) are associated. This could be caused by the circumstance that we need more words to express affects. Or: that higher educated persons (all participants are university students) express their affects with various words/ phrases whereas persons with lower education express their affects often with the same words (and show lower lexical entropy).
- Limitations P16, L 676: Maybe the education affects the ASI. Furthermore, you did not compare texts generated under the conditions positive versus negative vales (e.g. talking about a positive experience versus negative experience). In the meaning of criterion validity the ASI should be higher in the positive condition.
References
- There are too much self-citations, especially from Mr Salvatore.
Aafjes-van Doorn, K., Porcerelli, J., & Müller-Frommeyer, L. C. (2020). Language style matching in psychotherapy: An implicit aspect of alliance. Journal of Counseling Psychology, 67(4), 509-522. doi:10.1037/cou0000433
Pennebaker, J. W., Boyd, R. L., Jordan, K., & Blackburn, K. (2015). The development and psychometric properties of LIWC2015. Austin, TX: University of Texas at Austin.
Wiltshire, T. J., Philipsen, J. S., Trasmundi, S. B., Jensen, T. W., & Steffensen, S. V. (2020). Interpersonal Coordination Dynamics in Psychotherapy: A Systematic Review. Cognitive Therapy and Research, 44, 752-773. doi:10.1007/s10608-020-10106-3
Reviewer 3 Report
Affective Saturation Index. A lexical measure of affects
In my view, this is a tremendously interesting approach, and I also very much like the interdisciplinary nature of this research and paper. The latter makes the paper a bit hard to digest for me in some places, but I nevertheless very much appreciate the authors' effort to put more than just one strands of expertise together.
Intro: Reporting of the background is solid.
Methods: While much information on ASI is given, I ask the authors to provide the details of its computation in a fashion that allows other researchers to calculate it exactly as the authors did. I am sure the authors will find an appropriate way to do so. I think it is also in the interest of the authors, if other researchers replicate their findings or may corroborate the validity of this very index in other research designs, other populations, etc. Please do not only refer to other papers, where some steps of the calculation might be found, but information is provided in unsufficient detail, either.
rMSSD is a good choice for the HRV variable. Of course, the condition in which it was measured is not ideal, but the authors acknowledge this limitation in the discussion. I think, it can be accepted as a first approximation. Further research can dissect the relationship between ASI and physiological variables in more detail.
Line 520: Please exchange "Enter method" for "Standard Multiple Regression". "Enter" is just used in SPSS and may be other statistics software, but is not a correct statistical term.
Round 2
Reviewer 1 Report
To the point:
The abstract: "and FVI; AdjR2=.428 and AdjR2=.241, respectively) and the way the text's affective saturation reflects the intensity of the individual's affective state (HRV; AdjR2=.428)"
At this point, I have no idea what the "ADJ R2" is.
Furthermore, in comments about Tables 7 and 9 there is no explanation about that. At this point, I do not consider it clear the adjustment and why it was made.
Another significant point to be cleared is the statistical test. You wrote: "HP1 was tested by means of a regression model having the index of saturation (SEI) as dependent variable and ASI as a predictor." Then, the test is made using ASI and Words, but is it the same model applied to different sets? Or are there two models? Is there any sense in applying one model created based on ASI to Words in the first case? In the second case: there another model should be shown and analyzed.
Also, the positive/negative influences were not considered to the hypotheses stated. Why? Considering only one looks like it ignores that possibility and does not open to analyze why that happens. Is it important to the area?
The authors stated descriptive statistics as "technical details - we do not believe they need to be addressed with specific comments". So, why should I read it?
"mother-tongue" -- would be better to use "mother language".
"... or if they reported psychopathological symptoms over the threshold of clinical relevance." Be clear about the threshold (how is it defined? and why was it?)
Author Response
please, see the attachment

Reviewer 2 Report
Good work!
Author Response
Thank you for your positive feedback!
Round 3
Reviewer 1 Report
I tried to suggest improvements. I am not opposed to publishing the article as it is.